# Synanthropic Flies—A Review Including How They Obtain Nutrients, along with Pathogens, Store Them in the Crop and Mechanisms of Transmission

**DOI:** 10.3390/insects13090776

**Published:** 2022-08-27

**Authors:** John G. Stoffolano

**Affiliations:** Stockbridge School of Agriculture, University of Massachusetts, Amherst, MA 01003, USA; stoff@umass.edu

**Keywords:** dipteran diverticulated crop, bio-enhanced pathogen transmission, regurgitation, horizontal transmission of resistance, bats, birds, apes, zoonoses, viruses, emerging infectious diseases

## Abstract

**Simple Summary:**

The significance of non-blood feeding cyclorrhaphan flies with some synanthropic, dipteran families (i.e., Calliphoridae, Sarcophagidae, and Muscidae) in transmitting pathogens to humans and their food sources needs a lot of focused research. This review is designed to provide information on how these flies obtain nutrients and pathogens, store these, and then transmit pathogens to new hosts. Most of the studies in the literature have made references to mechanical transmission by these flies with some references mentioning their gut, and a few specifically mentioning the diverticulated crop organ involvement in the carriage of pathogens. New studies, where pathogens have been shown to multiply within the crop storage organ and, also where the crop is the main site for horizontal transmission of pathogen resistance, make the crop a bio-enhanced organ worthy of consideration in future epidemiological studies and models. Emphasis is also placed on nutrient fly resources in nature, such as bat excreta, enabling these flies to obtain additional pathogens.

**Abstract:**

An attempt has been made to provide a broad review of synanthropic flies and, not just a survey of their involvement in human pathogen transmission. It also emphasizes that the crop organ of calliphorids, sarcophagids, and muscids was an evolutionary development and has served and assisted non-blood feeding flies in obtaining food, as well as pathogens, prior to the origin of humans. Insects are believed to be present on earth about 400 million years ago (MYA). Thus, prior to the origin of primates, there was adequate time for these flies to become associated with various animals and to serve as important transmitters of pathogens associated with them prior to the advent of early hominids and modern humans. Through the process of fly crop regurgitation, numerous pathogens are still readily being made available to primates and other animals. Several studies using invertebrate-derived DNA = iDNA meta-techniques have been able to identify, not only the source the fly had fed on, but also if it had fed on their feces or the animal’s body fluids. Since these flies are known to feed on both vertebrate fluids (i.e., from wounds, saliva, mucus, or tears), as well as those of other animals, and their feces, identification of the reservoir host, amplification hosts, and associated pathogens is essential in identifying emerging infectious diseases. New molecular tools, along with a focus on the crop, and what is in it, should provide a better understanding and development of whether these flies are involved in emerging infectious diseases. If so, epidemiological models in the future might be better at predicting future epidemics or pandemics.

## 1. Introduction

As an anthropocentric culture, we tend to always look to the future, forgetting the past. Yet, early modern plague writers noted that one of the main organisms associated with impending plagues was flies [1]. Recently, Balaraman et al. [2] showed that SARS-CoV-2 could be mechanically transmitted by adult house flies, and this suggests that house flies might be more important in pandemics than previously thought. At the same time, current research efforts to identify causes or factors involved in predicting various emerging infectious diseases generally consider insect vectors as important and, presumably, they are referring to blood-sucking flies [3,4], but they usually ignored transmission by synanthropic flies. How have synanthropic flies been ignored when recent studies (e.g., fly involvement in anthrax transmission) show they are extremely important as transmitters of pathogens? The famous surrealist painter, Salvador Dali, must have encountered synanthropic flies and even considered them important because he did some wonderful paintings of calliphorid adults [5]. 

This review will attempt to influence your thinking about synanthropic flies in the lives of non-humans, as well as flies in early primate species. “It is likely that during centuries and until recently, the main route of simian pathogen transmission to humans was NHP (sic, non-human primates) hunting and wild meat consumption,” [6]. In his popular book (i.e., Spillover), Quammen [7] states “It (sic, a pathogen) can travel from one farm to another on a humid breeze”. More likely, the pathogen he is talking about can go from farm to farm via flying synanthropic flies. We cannot fail to recognize the importance and the involvement of arthropods, especially the 3 cyclorrhaphan families discussed here, as potential transmitters of pathogens of non-human primates (NHP), humans, domestic animals and wildlife. Furthermore, these flies can serve as “fly driven contamination” sources or what Hugh-Jones and de Vos [8] called a “multiplier hypothesis” whereby the flies can rapidly expand the size of the epizootic. In their review of the links and interfaces amongst wildlife, livestock, and human disease control, the authors [9] only mentioned mosquitoes and ticks, ignoring some significant non-bloodfeeding Diptera discussed here. 

Members of some countries often fail to realize that, in many countries, food is scarce, especially at certain times of the year. It is this scarcity of food or lack of food security that leads people to seek wild meat, whether it is hunted, found dead, or obtained somewhere in the world in a wet market. The risk of getting a pathogen is often not even considered. Finding something to eat is more important. This is what happened in 2011 in Chama, Zambia, when there were 511 human cases of anthrax (*Bacillus anthracis*) infections resulting in 5 deaths because individuals ate meat from a dead hippopotamus [10]. In addition to chronic food security issues, cultural habits often come into play. This happened in the spillover from the chimpanzee SIV to human HIV [11] and is believed to have involved the cultural habit of eating bushmeat [12,13,14]. In some countries, however, we should not dismiss the possibility of fly-borne pathogens being transported to the foods eaten [15]. More will be said about this later.

What Devaux et al. [6] said in his above statement still leaves unanswered the question ‘Where do NHPs or other non-human animals, such as bats and birds, that are often eaten because of hunger or cultural/religious reasons obtain their microbes/parasites. Eating contaminated foods could lead to subsequent human infections, epidemics and even pandemics?’ In an interesting paper, Bennett et al. [16], studying fruit bats, made the statement that there was a strong relationship between bats and arthropods. Important between this association is that bats are reported to house more zoonotic pathogens than all known species of mammals [17]. This association with arthropods was also emphasized in the paper by Gessain [18], who stated “Most of the persons included in this study were hunters of such NHPs, thus at high risk of contact with infected body fluids (blood, saliva, etc.) during hunting activities.” It is these dead animals, or their feces, that non-blood sucking flies of the calliphorid, sarcophagid, and muscid families are attracted to for egg laying and feeding on the blood and other body fluids, which they immediately dispatch into their crop organ and eventually use them for their own reproductive and/or metabolic strategies [19,20]. Once fed, and when flies find another food source, or place to lay eggs, the crop fluids are either externally removed by regurgitation or bubbling or sent internally to the midgut for digestion. If the midgut route is taken, the pathogens exit by defecation. Both processes (oral regurgitation and defecation) might contain a pathogen/parasite infectious source. This is probably the scenario of what happened in the study by Echeverria et al. [21] where they showed that adult flies, including house flies, accounted for numerous enteric pathogens causing diarrhea in the village of Ban Pong, Thailand. Caution to readers when seeing the term excretion in the literature because some authors use this term to define any product that exits either the mouth or anus of a fly. In this paper, the term excretion is not used, but emphasis is placed on regurgitation from the mouth and defecation from the anus.

In the major review by Devaux et al. [6], various ways are discussed in how cross-infection between NHPs, and humans might take place. In fact, if you also read the excellent books by Quammen [7] and Oldstone [22], you will notice that there is no mention of the 3 dipteran families presented in this review as to their involvement in any zoonotic diseases. It is estimated that 61% of known human diseases originate from wild animals [23]. Flies feed on their feces, wounds, and other secretions and may be involved in the transmission of pathogens between humans and wildlife. Sampling a known wildlife host (i.e., Indian Flying Fox) for zoonotic pathogens and using PCR techniques, plus family-level primers, the authors [24] estimated a minimum of 320,000 viruses yet to be discovered in various mammalian hosts. Non-blood feeding flies were not even considered in these previously mentioned references. Förster et al. [25] considered their study to be the first to demonstrate the potential of synanthropic flies involved in pathogen transfer. They analyzed 12 species of flies and found 24 species of bacteria and 2 species of fungi from various animal facilities sampled. Synanthropic flies have been dismissed and not even considered until recently in various zoonoses. Yet, the majority of emerging zoonoses are generally caused by viruses [24]. What is often ignored is what is called anthropozoonotic transmission that goes from human to animals and this reverse zoonoses has already been demonstrated with COVID-19 outbreaks on mink farms [17]. Involvement of insects in this transmission process has yet to be demonstrated. Even though the involvement of viruses has been demonstrated in some emerging infectious diseases of humans and animals, a considerable amount of research has yet to be done to involve synanthropic flies as transmitters of viruses. The focus of the above references and, the study of Förster et al. [25] leaves room for the ideas presented in this paper (i.e., three families of non-blood feeding flies of the Sarcophagidae, Calliphoridae, and Muscidae as pathogen carriers). Their consumption of nutrients obtained from various sources such as blood, feces, wounds, and other fluids from infected animals or the environment could contain pathogens or parasites that then go into the crop of the fly where they are stored. These pathogens could very well be obtained from numerous wild animals in human environments. It is this wild, animal–human–environment interface that is becoming so important in emerging infectious disease outbreaks.

The dipteran, diverticulated crop, for many years, was a forgotten organ from an entomological point of view [26] but has been neglected for its importance in most epidemiological studies. Thus, just like the crop organ, this group of flies has also been neglected or forgotten and should be considered a major item in future epidemiological models, especially when it comes to NHPs. Support for this is the study by Gogarten et al. [27], using PCR, obtained positively infected flies associated with mangabeys and suggested that *Bacillus cereus* Biovar *anthracis* (Bcbva), causing sylvatic anthrax, and *Treponema pallidum pertenue*, yaws, transmitted by flies. An attempt is made here to convince you that the fly crop can contain a deadly source of infectious pathogens. Its involvement in diseases is not purely mechanical, but regurgitation is a bio-enhanced transmission process. In the material that follows, a case will be made for the importance of the fly crop as a major neglected source of stored pathogens, and supporting evidence will be presented where these fly carriers could infect either fruits, leaves, or other foods of diverse animals. This pathogen transfer can occur either by an infectious fly being eaten or by the fly leaving its newly acquired pathogens/parasites via the processes of regurgitation and/or defecation on the food source. 

Cerretti et al. [28] provide approximate dates for the origin of the 3 dipteran families emphasized in this study, the evolutionary history of which is part of the focus of this review. Calliphoridae (blow-flies, carrion flies, ca. 34–23 MYA), Sarcophagidae (flesh flies, ca. 34–23 MYA) and Muscidae (*Musca* sp. and other synanthropic flies, ca. 47 MYA) are members of the insect order Diptera. They are mostly non-blood sucking flies often synanthropic and have been well studied with respect to their importance to humans in food safety [29], forensic entomology [30], potential vectors of human pathogens [31,32], and model systems for human physiology and genetics of feeding regulation [33]. What is needed, however, are research investigations demonstrating the role of these three families as transmitters acquiring potentially infectious pathogens from NHPs and, as contamination agents of the food of NHPs, bats, other wild animals, and humans in their involvement in any potential spillovers. Calisher et al. [34] reviewed the presence of bats as significant reservoir hosts of various viruses in emerging infectious diseases, however, he did not implicate flies as important in pathogen transfer.

It should be obvious that these dipteran families were present long before human primates originated (6–7 MYA) and, well before (32–47 MYA) NHPs (55 MYA) were present. These predicted timeframes give sufficient time for the development of evolutionary and ecological interactions, pathogen mutations, and adaptations between the fly associated pathogens, NHPs and other animals. At these earliest points in time, these fly families were not synanthropic, but maintained a close relationship with non-human animals. They obtained their essential nutrients by feeding on materials, such as feces, saliva, mucus, blood, and nasal secretions. It is true that these dipteran families are generalists; however, one must ask what bonds synanthropic flies to a specific animal group such as NHP and humans. For these flies to successfully exist, they must have nutrients for general maintenance and for reproductive events. These nutrients must be in quantity sufficient for population survival. To accomplish this, these flies should focus on animals that produce large amounts of both nutrients. 

Based on fossil evidence, there is little doubt that these 3 dipteran families were present as part of the larger fly clade known as the Calyptratae (67.5 MYA). In comparison, primates are believed to have originated as early as 65 MYA in Asia, not Africa, while *Homo sapiens* had its origin in Africa only 0.2 MYA. The animal reported to be in the main line of human evolution is *Archicebus achilles*, and is believed to be an insectivorous species that lived in the early Eocene Forest about 55 MYA. Evidence places the divergence between old-world monkeys and apes at 25 MYA during the Oligocene [35]. Did these early NHPs obtain sources of infection from flies in some way? Some studies have reported NHP where a potential host becomes infected by consuming infected dipterans or the vegetation they soiled via regurgitation and/or defecation. This option is supported by the study of Evans et al. [36] where they evaluated discarded plant material for viruses that were eaten by mountain gorillas and golden monkeys. They detected several viruses that were shed orally. Such discarded food items containing infective viruses could be picked up by flies seeking water and or carbohydrates and stored in their crops. They then could become what Cirillo [37] called “winged sponges”. Very few studies have focused on showing that members of these three dipteran families feed on bat feces or urine. More studies need to be done in this research area. Without a doubt, a restructuring of research directions needs to take a new view of the importance of this group of flies in future epidemiological models.

Greenberg [31] stated “What prehistoric man thought about flies will never be known”. What is even more significant in today’s theater of zoonoses is what types of interactions took place between NHPs, other animals, and non-blood feeding dipterans prior to the arrival of hominids and *Homo sapiens*? Can we get some idea(s) from current or planned studies? The interaction between these early, insectivorous species probably involved flies feeding on the feces of the insectivores and their various body fluids (i.e., wounds, saliva, mucus, or tears), whether they were alive or dead. Did these flies seek teary secretions from the eyes of insectivore species, as *Musca sorbens* and *M. domestica* now seek on humans? These adult flies take eye secretions, plus the pathogen *Chlamydia trachomatis*, into their crop. Once they feed on another animal’s eye, they can transmit the pathogen to another host via regurgitation. A recent laboratory study [38] showed that adult *M. domestica* (also incriminated in the spread of the trachoma pathogen) can take up the pathogen and put it into the crop where it remains viable for several days. Surely enough time to infect, via regurgitation, a new host. A similar situation exists for adult house flies obtaining spores of the anthrax pathogen from environmental reservoirs and storing them in the midgut. It is within the crop where they receive protection from various detrimental hazards until they are deposited in the environment via regurgitation/defecation for reuptake by a possible uninfected host [39]. Otake et al. [40] reported that Benfield and others (1992) noted that “The fact that PRRSV was more readily detected within the intestinal tract of the housefly was because this may enhance its ability to survive, because within the gut the virus may be protected from environmental factors such as exposure to ultraviolet light and drying”.

Identification of close relatives of the human adenoviruses in great apes of Africa suggests a zoonotic origin [41]. Analysis of 800 fecal samples showed that 55% of gorillas and 25% of chimpanzees carried the virus and Bayesian ancestral host reconstruction revealed 10 events during 4.5 MYA. Furthermore, they suggested that fecal–oral cross hominine transmission took place. Since I will discuss how synanthropic flies are involved in obtaining pathogens from fecal samples, it is quite possible feces were also involved in these transmissions. Hoppe et al. [41] also determined that more than 100,000 years ago, 2 independent transmission events occurred resulting in human infection.

Nutrients in the environment, or fomites, serve as a major link between infective stages of pathogens and a fly carrier. Since insects, including flies, cannot produce their own sterols, they must obtain them from plant phytosterols or animal sterols from feces. Research by Sistiaga et al. [42,43] showed that sterols are present in the feces of NHP, as well as Neanderthals (Middle Palaeolithic site dating to ca 50,000 years ago), which would have provided flies their essential exogenous sterols prior to the arrival of *Homo sapiens*. Nutrients (i.e., from feces, blood, or other secretions) obtained from early NHPs, as they are today, were and still are used by flies for inducing mating in some species and initiating egg development or vitellogenesis. While imbibing these nutrients, they also acquired numerous shed microbes or parasites. For general body metabolism, maintenance, and flight, in addition to sterols, it has always been essential for flies to obtain carbohydrates in the form of honeydew, nectar, exudates or juices from fruits and some plants [44,45,46]. The importance of fruits in epidemiological models cannot be overemphasized. In fact, Maldonado and Centeno [47] include fruits as one of the important parameters in developing the danger index for synanthropic flies, especially as they noted that most fruits are usually eaten uncooked and without washing.

There is very little in the literature as to whether cyclorrhaphan flies, other than the drosophilid fruit flies, feed on fruits for their carbohydrates and, at the same time, defecate on them leaving behind numerous pathogens. This information is needed, especially in the tropics where animals may feed on soiled fruits and acquire pathogens from the flies that have regurgitated or defecated on the same fruits. An example of the isolation of a virus from a piece of fruit, presumably eaten by an infected bat was demonstrated by Chua et al. [48]. It is this nutritional drive to stay alive and reproduce that puts these dipterans into the epidemiological cycle of numerous diseases (zoonoses) and collision with both NHPs, many other animals, and primates. 

It is estimated that there are 505 different species of NHP living today, mainly in 4 nations, which are reported to harbor 65% of all primate species (i.e., Brazil, Madagascar, Indonesia, and the Democratic Republic of the Congo). We know very little about the deaths of these primates since bodies are not usually found. What is needed is some estimate in various areas as to how many deaths occur and, how many NHPs and other animals are available as nutrient sources of flies feeding on their feces, bodily secretions, and the amount of biomass available to support fly larvae when these animals die. As some animals, including NHP groups, face extinction, what effect will this have on the flies? Olson et al. [49] provides an excellent review concerning evaluating various fecal sampling designs for determining the presence of feces and density of producer, using gorilla feces in the Republic of Congo as an example. Such a sampling strategy to determine what primates and other animals are feeding on will provide needed information on which organisms are struggling for survival. Most of the literature on NHPs has focused on larger primates, such as gorillas and monkeys, which leaves many NHPs for which their fly associations remain unknown. The extinction of any primate species that is involved in fly survival could force flies to seek other NHPs for survival. What a treasure for researchers to explore.

The evolution of flight was significant to insects, bats, and birds, all of which serve as food sources for one another and, all of which can share novel viruses, microbes and parasites acquired with their food. A review worth reading is the paper by Chan et al. [50] who reviewed the literature on bats and birds and their involvement in interspecies transmission of novel viruses. Wherever there are microbes and parasites, there usually are flies; and, since adult dipterans can fly [51], they have direct access to numerous pathogens by eating from the various infective substances produced by bats, birds, and other animals. Thus, all three flying animals, especially flies, pick up and become flying agents or ‘sponges’, according to Cirillo [37], and act as dispersal agents to and between NHPs and other animals. These three families of dipterans have been ignored in most epidemiological studies looking for ‘spillovers’ into the numerous zoonotic zones, especially in tropical Africa and Asia. These three dipteran families have a predisposition to feed on and breed in rotten vegetation, carcasses, and feces, plus other secretions of many NHP species. Out of a total of 1037 flies collected and identified from a survey of garbage, kitchens, and cafeteria sites in Malaysia, members of all three families discussed here were identified [52]. It is well known that the adults of these three fly families obtain numerous pathogens and parasites while feeding on the previously mentioned products coming from many animal species [53]. 

The major method of picking up and transferring various infectious agents while feeding has been mainly attributed to numerous studies demonstrating that their transfer method is a mechanical process. Several reports note that the infectious agents are picked up by the legs, mouthparts, or wings while grooming, and then passed on to their next food source [54,55]. Based on these early studies, these flies were classified as major, non-blood feeding, mechanical transmitters of human pathogens. The excellent 2-volume series by Greenberg [31] is a major resource for those interested in the biology of flies and the numerous pathogens and parasites, which they can mechanically transmit to other organisms, including humans. Caution, however, is that since 1971, the literature and significance of non-blood sucking flies as transmitters of human and domestic animals has exploded [51]. In many cases, this transfer involves more than mechanical transmission via external structures, which also includes the crop organ. An example is the early review by Carn [56] where the author states, “Many different species of Diptera are implicated in the mechanical transmission, but haematophagous species are the most important”. Hopefully, this review will help dispel this statement. 

## 2. What Is a Synanthropic Fly? 

These undomesticated flies live in close association with and, in some way benefit from the association with humans. The literature reveals that almost all studies on synanthropic flies focus on a survey of flies associated with humans, the pathogens they carry, types of diseases associated with them, and the type of traps used to collect them. No paper was found on behavioral or physiological mechanisms that might connect these flies with humans or their domesticated animals.

The one physiological/behavior mechanism that might connect a fly species with humans is the ability of house flies to detect lactose. It is suggested that the common house fly, *M. domestica*, evolved the ability to taste and respond to lactose while other flies do not have this ability, thus the house fly’s association with lactating animals. Instead of the smaller house fly, it is the larger calliphorids or sarcophagids [57] that make better transfer agents. In the paper by Maldonado and Centeno [47], the authors modified Mihályi’s danger index [58] to include synanthropic information permitting them to discriminate between flies that were asynanthropic, hemisynanthropic, and eusynanthropic (Figure 1). In that same paper, the authors state, “The idea of a direct relationship between body size and the capability of contamination and transmission of pathogens seems to be supported by several researchers around the world (Mariluis et al., 1989, Brown 1997, Graczyk et al., 1999, 2000, Fischer et al., 2001)”. If size is important, then for sure, it involves the crop organ. The bigger the fly, the more the crop can hold, and it holds more than the midgut. Mihályi [58] did not consider the synanthropic index (SI) sufficiently informative, in a sanitary sense, and for this reason he made up a danger index keeping in mind 3 aspects of the behavioral and morphological characteristics of the insects: the undergoing infection, the passing infection and body size of the fly transmitter. Another important paper is that of Blacio et al. [59], where they collected 2925 specimens of Calyptratae, but surprisingly, they did not collect any house flies. Nuroteva [57] is an important paper to read where the author determines a synanthropic index for flies in Finland and provides a broad analysis of papers prior to his publication as to how different regions show different SI for the same species. Regardless of which synanthropic index is used to calculate which flies or animals are found in association with humans, more studies need to be done in areas where flies have proven to be involved in transmitting human or animal pathogens, such as anthrax or food-borne pathogens. 

## 3. Spillovers and Flies

It is significant to note that in their extensive review, Morse et al. [4] found that of the 868 species of pathogens identified as causing human diseases, 61% were zoonotic. Since flies need external sterols, it goes without question that they obtained them from wild animals or possibly plants. These flies obtain their main nutrient, protein-based diet by feeding on various products of wild animals (i.e., such as feces). Furthermore, as humans invaded the habitats of wild animals, they certainly would come to also share their pathogens.

Preparing for the 2020 spring break recess at the Univ. of Massachusetts in Amherst, which occurred during the COVID-19 outbreak, I asked my colleague Dr. Stephen Rich to recommend a book on zoonoses that I could read during the break. Dr. Rich was the lead author [60], and was also instrumental, in the study showing that there was a spillover years ago from chimpanzees to humans of the Plasmodium currently causing malaria in humans. The later study of Duval et al. [61] suggests that the spillover was from gorillas. The book Dr. Rich recommended was *Spillover: Animal Infections and the Next Human Pandemic* by Quammen [7]. It was a significant read, forcing me to really think about what was going on with respect to flies of these three families and NHPs in tropical areas, especially Africa. How were these NHPs infected and by what method did they obtain their infectious source? Were non-blood feeding flies somehow involved? Because of the enormity of this topic and, the proliferation of literature in this area, I might have missed an important paper and apologize to the author(s) and reader(s). As Quammen [7] said, “We (sic all organisms) share this place we call earth”. Consequently, all organisms are striving to succeed and capture the essential elements needed to survive, such as food nutrients. Since flies cannot make their own essential sterols, they rely on materials (e.g., feces) mainly from other animals and possibly plants. Thus, we should view all microorganisms (i.e., bacteria, fungi, and viruses, plus parasites) as potential competitors and realize that the impact of humans moving into previously, what we could call vacant human environments, has impacted the environment but has also brought us even closer to many of these pathogenic agents.

Throughout my career, I have studied the biology of non-biting Diptera. One of my major research areas is the significance of the ventral, diverticulated crop of adult dipterans. My laboratory group has studied the various mechanisms involved in crop regulation [26], the significance of the crop in feeding, and most importantly why flies are flying storage tanks or sponges and the importance of what is in these tanks or on the sponges, which are various microorganisms [51]. Before beginning this journey into the significance of the crop as a major storage organ of numerous infective agents, I had little knowledge of how many reports presented in the literature and are now cited here, involve the crop as a storage organ for various pathogens. I am sure I missed some and apologize to the authors. Since several studies have shown that non-biting flies can act as mechanical transmitters of human pathogens [62], here we will focus on the importance of the crop and how the crop of non-biting flies serves as an important storage vehicle for numerous pathogens and some parasites of NHPs, humans and other animals. Since the literature may be scarce on the interactions between the three fly families and other animals, I will also include important literature sources on various animal species, such as bats.

## 4. Were Synanthropic Flies Originally Associated with NHP and Other Animals? 

I searched for a term similar to synanthropic but refers to the fly association with NHP and early hominid species with little success. Based on the analysis of projected origin dates (house flies, 86–48 MYA; NHP, 85–55 MYA; hominids, 14 MYA), it is obvious that the three fly families (i.e., Calliphoridae, Sarcophagidae, and Muscidae) were present prior to the evolution of both hominids and humans, as we know them. However, based on when they are believed to have evolved, it is also obvious that house flies and NHP occurred at about the same time. Spillar [63] examined the diets of some muscoid flies and noted that for some larvae, the importance of lactose had not been examined, yet is essential to lactating animals. The importance of lactose for the now common, house fly, *M. domestica*, however, would change possibly making it a very synanthropic species. Schnuch and Seebauer [64], based on electrophysiological experiments of taste chemoreceptors, showed that adult house flies can taste lactose, while experiments with other flies failed to show a tarsal or labellar chemoreceptive response to lactose. The ability to taste lactose might have been a key factor in the co-evolution between the house fly and humans in the highly domesticated situation of humans using lactating animals for milk. However, house flies originally had close contact with groups of lactating animals and, only later after a gradual transition, did they maintain a close relationship with humans. Is there any reported evidence of adult house flies feeding on the lactose-based secretions of NHPs? Wiesmann [65] examined the crops of field-caught house flies in stables and farmhouses and showed that the crops all contained lactose. Thus, a new food source for adult house flies. According to Wiesmann [65], *Musca domestica* ignores the nectar of flowers as an energy source, but searches for its food in human habitations and in stables of domestic animals [65]. The literature is scarce as to house flies feeding on nectar or honeydew. What does house fly feed on in areas lacking or deficient in domesticated animals and, are the chemoreceptors of adult house flies in those areas still able to detect lactose? It is suggested that lactation may have originated from pre-mammal ancestors over 150 MYA [66]. It is proposed that, in adaptation to the availability of milk in its environment, *Musca domestica* acquired the ability to perceive, as well as to metabolize, lactose. More research needs to focus on the behavioral and physiological mechanisms of synanthropic flies.

One could ask if non-bloodsucking flies ever show a close association with groups of NHP and, would this put them in a position to pick up and transfer pathogens they house? The answer is yes, but a lot more research needs to be done. “Primate-associated flies were observed to move between groups of different species, suggesting that they could be involved in transmitting the yaws pathogen between species, even when NHPs are not found in mixed-species associations” [27]. A recent report by Gogarten [67] notes that by studying the fly and monkey associations that larger monkey groups harbored higher fly densities. Figure 2. shows the close association between flies and a sleeping chimpanzee. Gogarten’s paper [67] is a must-read if one wants to know more about whether NHPs have developed defensive behaviors against nuisance flies and what we should call this close association with NHP.

We can also imagine that many of the current and known pathogens are being moved around amongst NHPs and other animals, including bats (50–60 MYA) and birds (modern birds, 150 MYA; [68]). Did house flies co-evolve with humans, as we know them, or was it present for interaction with NHPs? As previously stated, the house fly is reported to have evolved 48–86 MYA while *Homo sapiens*, as we know, entered the picture only 0.2 MYA. An important question is, what did *M. domestica* feed on as adults prior to the presence of *Homo sapiens* and, the use of milk from domestic animals? Free lactose is only present in lactating mammals and lactating animals had their origins in a pre-mammalian ancestor over 150 MYA [66], which is a much longer period before the house fly is reported to evolve. 

In an important study, Gogarten et al. [27] examined, marked, and recaptured flies that remained with the monkey troops for 12 days and for up to 1.3 km of their movement. Of these flies, 45.8% were shown to be Muscidae, 35.4% Calliphoridae, and 8.3% Sarcophagidae. Sixty percent of the flies captured in feces were Muscidae. Can this behavior of flies following monkey troops be considered a forerunner of synanthropic behavior? It is well established that synanthropic flies are those flies that are closely associated with humans, as we know them. What do we know about the flies associated with other hominids, such as Neanderthal, Denisovans, and more recently the ghost hominid? Why did *M. domestica* become the only truly synanthropic, non-blood feeding fly remains an academic issue but, which probably could be solved by doing gene mining and asking when the lactose receptor evolved in house flies? 

Another question that needs to be answered is, which fly species contribute most to transmitting various pathogens? If you ask someone interested in food safety or domestic animal health, the answer would universally be the common house fly [69]. Historically, and even to date, more research has been devoted to this fly because of its close association with humans, our domestic animals, and its feeding on the numerous types of filth associated with humans and domestic animals. An interesting and a must-read paper is by Maldonado and Centeno [47] who attempted to answer the above question. They analyzed data for seven Calliphorid blowflies and the adult house fly to produce a synanthropic or danger index. They mention that previous workers proposed there was “… a direct relationship between body size and the capability of contamination and transmission of pathogens seem to be supported by several researchers around the world.” Their results did show that size was significantly important in having a high danger index. This might make sense if one considers the size of the crop of larger species and the number of infectious agents it could store. In addition to crop size, however, the rate of regurgitation should also be considered. Maldonado and Centeno [47] proceeded, and based on their analysis, they concluded that 5 of the calliphorid blowflies were better suited as pathogen transfer agents over the house fly and, only one blowfly was lesser of a problem. This may seem like a surprising result but, is the house fly a major disseminator of pathogens in tropical rain forests? House fly is rarely reported in tropical rainforest settings, especially with NHP. 

## 5. Meal Analysis of Filth, Necrophilic and Synanthropic Flies

Several studies have reported where meal analysis of collected filth flies was obtained and science has given us several new tools and a new technology (i.e., using non-blood feeding flies as biosentinels) to examine feeding habitats, for the type of food (i.e., feces, bodily fluids) the fly has fed upon, but also both the species name of the fly involved and identification of the various pathogens/parasites the meal harbored [27,53,70]. The paper of Owings [53] should provide the reader with references relating to this topic without having to reference them here. Observations have also been made that fly feeding on wounds can also be involved in transmitting pathogens [71]. Unfortunately, some of these excellent studies, for various reasons, did not identify the species of fly [70], while most did not locate within the fly’s body where the main source of the meal was located. They also did not identify whether the infective agent was external or internal to the digestive tract. Information on the route of pathogen transfer to new hosts is needed (i.e., mechanical, oral, surface, or anal route).

*P. regina* can hold up to 2 µL of liquid in its midgut and 18 µL within the crop [72]. Thus, the crop probably is the major source of identifiable pathogens/parasites in these previously mentioned studies. This was shown in the study by Satchell and Harrison [71] who reported, and based on microscopic analysis, that the infectious agent *Treponema pallidum* subspecies *pertenue* (TPE), was mainly found within the crop and rarely in the rest of the gut. In a later review, Stamm [73] reported, “Although the early studies of Castellani and Lamborn showed that flies can act as a vector for yaws in an experimental setting, confirmation of the role of flies as a vector for yaws in a natural setting remained elusive.” However, molecular studies raised questions about the source of the fly-associated TP DNA found in baboons and humans. Stamm [73] concluded that the use of molecular tools and epidemiological methods is needed to specifically elucidate the role of flies (mainly *M. domestica* and *M. sorbens*) as transfer agents of the yaws pathogen. 

Using molecular techniques, such as vertebrate DNA sequencing, to evaluate what filth flies had fed upon, most studies where dissections have also been done report only that the gut was removed and analyzed [53,74] with no reference to which part of the digestive tract (foregut including the crop, midgut, and/or hindgut) the infective source was located. This present review emphasizes that the crop should be the main source of identified food sources in most studies. The crop holds more fluids and usually contains the fluid the infective agents were living in, and they can be retained for many hours/days post-ingestion. In addition, they are subjected to less anti-pathogen response in the foregut than the midgut or hindgut. Calibeo-Hayes et al. [75] fed adult house flies fluids containing the turkey coronavirus and showed that the infective virus remained in the crop for 9 h. When they placed infected flies with uninfected turkeys, the birds became infected. Pileri and Mateu [76] reviewed the importance of adult house flies as vectors of porcine virus. They suggested that even though it has been demonstrated that flies can serve as transfer agents, they considered this mode of infection (i.e., regurgitation) as a minor mode of transmission. In another study, Brewer et al. [38], in a laboratory study, showed that when adult house flies were fed *Chlamydia trachomatis*, the pathogen remained viable within the crop for 24 h, enough time for the fly to go to an uninfected host, regurgitate into the eye fluids, and infect a new host. Again, these references demonstrate the potential importance of the crop in pathogen transmission. More information is needed on the crop volume for various species; and, based on this information, researchers might be able to get a better idea as to whether the dosage of pathogen found in one fly is enough to cause an infection. Of course, this will depend on the type of infectious agent being considered and the size of the crop. 

Kobayashi et al. [77] showed that enterohemorrhagic *Escherichia coli* 0517 proliferates in/on the mouthparts and they termed this as bio-enhanced transmission, rather than mechanical transmission. The interesting fact is that the proliferation of the pathogen on the mouth parts in their study may have really resulted from material that was constantly being released from the crop during bubbling or regurgitation and, consequently remained on the mouthparts forming a biofilm. The significance of biofilm formation to infectious diseases and, the importance of biofilms in the fly crop have been discussed elsewhere [51,78,79] suggesting that *E. coli* can multiply in house fly regurgitant, a potential source of infectious material. Because of these studies and the future importance of the crop, researchers should make certain whether the fly species they are working with is a bio-enhanced transmitter and, not just a mechanical transmitter. Thus, it cannot be overemphasized or ignored that the importance of the diverticulated crop must be considered in future epidemiological models and studies!

## 6. Are Female Flies Better Transmitters of Pathogens/Parasites Than Males? 

Research from my laboratory shows that both males and unmated, previtellogenic females of *P. regina* visit feces in equal numbers and that at carcasses, fewer males but mainly mated, gravid females ready to oviposit are present [19]. This seems to be the case for most calliphorid species; and, if this is the general behavior between males and females of other species, it is obvious that females would be the better transmitters of pathogens and/parasites. Chaiwong et al. [80] also found in their extensive collections that females were collected more than males at all study sites while Sasaki et al. [81] stated, “The frequency of excretion (sic, defecation) by females with developing eggs in their ovary was clearly higher (6.5 min/drop) than for males or females with mature eggs.” This frequency of excretion should enhance the possibility of females with developing eggs picking up pathogens or parasites more frequently from animal feces. Female *P. regina*, as other synanthropic flies, probably show up more in carcasses than in feces [19]. Figure 3 shows a dead chimpanzee and the number of flies already feeding on secretions and laying eggs. Sasaki et al. [81] reported female house flies produce more defecation spots than males while El-Bassiony and Stoffolano [82] reported that male house flies produced more regurgitation spots than females. Depending on whether the fly is an oral, surface, or an anal transmitter should influence which sex would provide the best data for regional surveys using flies as biosentinels for the presence of pathogens and parasites.

Braack and De Vos [83] showed that in trap collections, using rotten meat and fish, in Africa, they collected significantly more females of both *C. albiceps* and *C. marginalis* (total between the two was 25 males to 186 females). These results agree with those of Stoffolano et al. [19] who showed that more females were attracted to decaying animal products than males. To obtain the best vertebrate DNA detection, females were more useful than males [53], indicating that it is mainly the female that goes to the dead vertebrate host for ovipositioning. A female with developing eggs not only visits carcasses more frequently, but she also produces more regurgitation/defecation than females with eggs ready to be laid. Thus, these females make better anal disseminators of pathogens and parasites. If this male-to-female presence at carcasses holds true, it might be best for future biosentinel studies to focus on female flies. If, however, one is interested in pathogens and parasites obtained from feces, either sex would be adequate. 

## 7. The Role of Fly Regurgitation and Grooming in Pathogen Handling

Most reports discuss flies obtaining and disseminating infective agents by what they called mechanical transmission, but little consideration has been given to the fact that flies regurgitate/bubble and often spread regurgitant, plus remaining fluids, on their bodies in the grooming process. Thus, the fluids in the crop have now become part of the external surface if the fly grooms and spreads pathogens from crop fluids. In the review by Stoffolano [51], more is said about the importance of grooming and its impact on supplying crop contents to a host, which often contain infective agents released from the crop. Remember, the crop has been reported to be a safe haven for many pathogens and parasites because it usually contains the fluids the pathogens were previously surviving and multiplying. If the grooming process is involved in transmitting a specific infectious agent, the importance of how long the agent can live on the outside of the fly must be considered. The most extensive study to date on grooming, in the three families of flies discussed, showed that flies were able to remove a considerable number of bacteria from their body via the process of grooming and that contaminated flies with bacteria were groomed more [84]. Authors suggested that for some bacteria the presence of bacterial attachments and possible stickiness facilitated them adhering to the body. What the authors could have done is to use SEM to check this and, to consider the possibility of antimicrobial peptides from the labellar and salivary glands in killing the bacteria, which would not have been detected in their Petri dish assays. Grooming behavior in flies, however, is important information for future epidemiological studies and more investigations are needed.

## 8. The Role of the Fly Crop, the Role It Plays Serving as a Storage Organ for Pathogens and Parasites That Are Both Obtained from an External Source, and Later Transferred to NHP and Other Animal Hosts

Traditionally, the crop of flies was considered mainly as a storage organ for carbohydrates (i.e., nectar, honeydew) [33,46] and proteinaceous nutrients that are difficult to obtain in the wild. It has also been shown to be a major organ for the storage of pathogens [51] while less is known about its ability to store parasites. The crop is an isolated foregut, diverticulated storage organ and it is within the crop that horizontal transmission of resistance to microbial antibiotics occurs [85,86,87]. A recent review of the crop in the genetic and molecular models as a major component of the gut of *Drosophila melanogaster* is available online [88]. 

The crop volume of different fly species is not well known. Laboratory studies with adult *M. domestica* and the blowfly, *Phormia regina*, have shown that the diverticulated crop is able to store considerable quantities of imbibed fluids (2–5 µL in house fly [89] to 18 µL in *Phormia regina* [20]), often containing infective doses of various pathogens or numbers of parasitic infective agents. When imbibed they in one way or another may be able to cause infections of human, domestic, and wild animals resulting in diseases (cholera, [90], *Escherichia coli* [79], *Chlamydia trachomatis*, [38], ORF of goats/sheep, [91] and others). If the fly fills its crop on infected fluids, it probably contains enough infectious agent material to infect another host if a single fly was eaten. Gogarten et al. [27] raised the same question as to whether a host can get enough infective agent (e.g., in his reference to Bcbva) to become infected, such as when monkeys or apes consume either flies or ‘vomit spots’. Gogarten (personal communication) noted that he has seen chimpanzees in the wild consume flies that they just caught in their hands. More information is needed on crop volumes of different species of flies, especially calliphorids, what dosage of an infectious agent they contained, and are these dosages enough to initiate an infection.

Several studies have shown that pathogens can also remain on the external surface of the fly for extended periods of time. What is new is that these reports were also able to show that the infectious agents can remain with the fly and even remain viable in their feces or regurgitant. The crop of non-biting flies is an important structure in disease transmission of both human and non-human pathogens and, most reports fail to consider its importance. Infective fluids picked up from a feeding source are stored in the crop, can be regurgitated, groomed over the body, and/or passed into the midgut for later digestion followed by defecation. 

## 9. Importance of the Fly Crop and Significance of Regurgitation/Defecation in Pathogen Transmission

There is little doubt that the dipteran crop, compared to other fly structures, is the major source of infective, regurgitant material containing numerous pathogens [26]. In their excellent paper on flies transmitting the pathogen of yaws, which is a major problem of wild baboons in Tanzania, Knauf et al. [92], using DNA sequencing, were able to show that several necrophagous flies contained the pathogens and were involved in the transmission cycle in the study area. What they failed to do, however, was examine the fly’s crop. Instead, they mentioned that the pathogen was probably transferred on the outside or exoskeleton of the fly. We now know better, and future studies need to look more closely at the fly to definitively know the site of pathogen presence.

Onwugamba [87] stated, “Flies primarily transmit bacteria through mouthparts and regurgitation and less likely through defeacation”. If this is true, the crop becomes an even more important organ in the epidemiology of diseases. Numerous studies have demonstrated that various pathogens have been ingested into the crop, and stored where they can remain for several hours, and even days. At the next feedings, the crop contents can be regurgitated onto substrates or substances (e.g., grass, fruits, spinach) that NHPs, humans and other animals (i.e., bats) might encounter and possibly eat [39,93,94,95]. It was also shown that a regurgitant droplet coming from the fly’s crop contained spores of *Bacillus anthracis* while feeding on anthrax-infected dead animals. In the case of the spores, they were retrieved from regurgitant droplets on or near the carcasses [94]. In an earlier study on the uptake and dissemination of anthrax spores, Braack and De Vos [83] injected radioactive phosphorous into the jugular vein of a male impala and 13 min later to make sure the injected material had circulated, they euthanized the animal and made a sizeable cut in the chest. This opening made an available source of body fluids, including blood, for two species of *Chrysomia* flies to obtain the labelled body fluids and then to leave the carcass to further process their meal. Non-blood sucking flies, unlike mosquitoes, do not rapidly eliminate the large volume of water in a meal via the anus to have an efficient flight. Rather, these flies find a suitable resting site where they eliminate water via regurgitation/bubbling and some defecation. Four days later, Braack and De Vos [83] were able to find what they called “fly droplets” on vegetation, very near the carcass. They found 19 fly droplets per leaf at a height between 1 and 3 m, just the preferred height that browsing kudu would feed and obtain the excreted spores. In their paper, the authors refer to the discarded fluids from the fly as “faecal droplets”, “discard droplets”, “fly droplets”, “Blowfly droplets”, and “infective droplets”. Probably most of these droplets were from regurgitation/bubbling behavior. In their paper, however, they made a major error. First, these flies are not hematophagous; and their attempt to equate the rapid excretion of water from the anus, as is found in blood-feeding flies, and which they called “discard droplets” to distinguish the anal secretions, which are darker, and more pasty was incorrect. Non-blood feeding flies do not rapidly excrete water via the anus as a clear liquid from the meal. Instead, they might excrete a clear-fluid droplet from their hindgut, but not from the midgut. It is unfortunate that this study did not include individual observations of flies to determine what was going on. Were these fluids coming from the regurgitation/bubbling or defecation process? This type of study is exactly what was done for four species of flies, two *Musca* sp. and two blowflies [82]. Results of their study showed that the number of elimination spots varied with each species and, the diet greatly influenced both regurgitation and defecation. Other studies have demonstrated that a pathogen, once ingested can still be detected 3 days in feces and can remain in the crop for 4 days [81]. Can NHP and other animals obtain pathogens from eating contaminated fruit or other types of foods? The main question here is whether bats, NHPs, other animals, or even humans can obtain pathogens from the regurgitant from the crop or feces of flies having ingested pathogens from an infectious source. 

## 10. The Ability of the Fly to Infect a New Host

Readers are urged to read the following paper by MacFarlane et al. [96] where the authors focus on the Asian–Australasian Region and summarize much of what current zoonoses and emerging infectious diseases are all about—“Humans create ecologically simplified landscapes that favour some wildlife species, but not others. Here, we explore the possibility that those species that tolerate or do well in human-modified environments, or ‘synanthropic’ species, are predominantly the hosts of zoonotic emerging and re-emerging infectious diseases (EIDs)”. Missing from the paper, however, are the names of the important flying transports, known as synanthropic flies, that carry on their bodies, within their crops, and hindguts infective fluids ready to be dropped on fruits, foods we eat, and possibly what we might drink. Are these wildlife species not the same animals we share our food supplies? Their study suggests that the most likely species involved in emerging infectious diseases are synanthropic species and, when they determined the ‘danger index’ or Mihályi’s index, they found that one of the most important measures was the size of the fly. This makes sense because larger flies have larger crops and midguts to store imbibed infective fluids. Thus, the larger the fly, the greater threat when it comes to emerging infectious diseases. Information about crop sizes of potentially important flies is lacking but is needed to better understand the importance of the crop in epidemiological models. In addition, the larger the fly, the more it can store in its crop and, based on the research of Rivers et al. [97], adult *Calliphora vicina* could retain evidence of blood or ingested semen 25 days post-infection for blood and 30 days for semen. This ability to retain fluids within the crop for extended periods should make larger flies better transmitters of pathogens. 

## 11. The Dipteran Crop as the Site for Horizontal Transmission of Resistant Genes, Possibly Mutations, and Reassortment of Genetic Material in Pathogen Spillovers

In their paper on the detection of antibiotic-resistant bacteria and multi-drug resistance genes from house flies, Akter et al. [98] report that house flies can disseminate numerous pathogens via “…regurgitation of gut contents…” with seven references to the gut and no mention of the crop. As previously discussed, regurgitation of any material comes from the crop, and not usually from the midgut in flies. Future studies must be careful in the terms that they use because they can often create misconceptions about the involvement of specific organs of the fly and their involvement as transmission vehicles. 

In a very interesting study involving several animals from pigs, cows, mice, and even flies, Marshall et al. [99] found that the wild-type *Escherichia coli*, which was bearing a transferable plasmid, was studied in a farm environment to track the movement of the bacterium amongst a group of organisms we could call farm animals. What they found was that the bacterium was spread amongst all the animals and was even found in the animal bedding. The species of fly was never determined in their study but assumed to be house flies. The following statement was most interesting, “The highest concentrations (sic, of bacterium) were recovered from horizontal wooden surfaces within the test pens…” This suggests to me, as a fly specialist, that these were surfaces where adult flies had rested to remove water from the crop and digest the meal. They regurgitated, defecated, and this caused these horizontal wooden surfaces to have the highest concentrations of bacteria. How often we neglect the smaller and seemingly unimportant parts of a zoonotic system (i.e., here the fly and its crop involvement). This study clearly demonstrates how a pathogen can move throughout a system and probably involves a small fly. In fact, horizontal transmission of antibacterial resistance has already been demonstrated in house fly [86,87,98,99]. Akhtar et al. [100] noted that “The rate of plasmid transfer was 10^−2^ (T/D) in the house fly midgut and 10^−3^ (T/D) in the crop.” This information should be taken seriously and should send a message to health workers to install up-to-date methods for controlling flies in hospitals, at markets, food preparation facilities, and even in the home. This is especially important in the increase of peripheral diabetic wounds that flies have been attracted to and shown to deposit pathogens. At the same time, the fly visitor regurgitates and defecates within the wound causing additional problems.

Where do resistant genes come from and what mechanisms do they employ? This is a topic beyond the scope of this review but, the reader is referred to two papers that provide adequate information about this topic [100,101]. This quote, “To better understand factors that promote the dissemination of resistance genes and to elucidate relationships between antibiotic resistance genes of producer, environmental, and pathogenic bacteria, new and improved strategies for sampling and screening of microbial populations and metagenomic libraries are needed”, [101], is appropriate here. One structure involved in the dissemination of resistant genes in the flying fly capsules is the crop. The importance of the crop as the site for horizontal transmission of resistant genes needs more research, especially when it comes to pathogens involved in zoonoses. 

Since the horizontal transfer of resistant genes in bacteria has been reported to occur within the crop of the fly, future studies should focus on the crop contents and the antimicrobial peptides from the salivary and labellar glands that are present when the fly takes its meal. One should also determine, using the in vitro, biological organ crop system to evaluate various aspects of resistance and what effect the fly’s own defenses might have on resistance mechanisms. The system reported by Wang et al. [79] provides procedures for using this system called the “crop vessel bioassay” to address some of these issues.

## 12. Do Non-Human Primates, or Other Animals, Feed on Insects and/or the Infectious Droplets of the 3 Families of Flies? 

“Disentangling the complexity of maintenance hosts or communities in multi-host systems at the wildlife/livestock/human interface is a difficult task” [102]. The disease ecology of a pathogen is complex and often flies are omitted as potential key players. In many tropical forest systems, duikers are often considered as key elements. They are browsers, thus eating vegetation that often contains contamination droplets on vegetation from flies that have just fed on an infected animal. Duikers follow both flocks of birds and troops of monkeys where they are reported to take advantage of fruit drop, which could be contaminated with pathogens excreted or regurgitated by flies or by frugivorous bats. Duikers are reported to eat insects, which could also include flies housing an infectious dose of pathogens in their crops. Thus, their eating habits include several different types of food that could be involved in a pathogen multi-host, maintenance system. It is now well established that flies can obtain a pathogen, such as *Bacillus anthracis*, while feeding on an infected dead animal and later regurgitate and defecate on vegetation that is subsequently eaten by an uninfected animal host [103]. The most extensive review on flies transmitting pathogens to fresh produce during field cultivation is that of Alegbeleye et al. [15]. Some specific examples are also provided in this review. Patrona et al. [95] also showed that the monkey pox virus could be found in both monkey feces and flies and, suggested that flies could possibly be involved in indirect transmission. The carryover from the behavior of *Archicebus achilles*, which is reported to be insectivorous, has not been lost in most non-human primates. Body size from the smallest non-human primates, which are insectivorous, to the largest (i.e., gorilla, chimpanzee, bonobo, orangutan, and gibbon) are mainly frugivorous and folivorous, but retain and have an insectivorous component to their diet. An evolutionary bit of evidence is the presence of CHIA or chitinase genes in all primates studied, thus providing evidence that most early primates were able to consume insects [104]. The authors go on to say that “Our findings are consistent with the hypothesis that primates produce acidic mammalian chitinase (AMCase) as a digestive enzyme for the breakdown of insect exoskeletons and that this enzyme is under more intense selection in more insectivorous and smaller-bodied primates.” In her book, Lesnik [105] mentions that of the primates that eat insects, it is the female that usually consumes more. Paleoanthropologists should also be looking into the issues presented in this review and their studies might provide more information as to how flies, prior to arrival of humans, are involved as transmitters of pathogens. 

Dead gorillas provide an enormous amount of food for both fly larvae and adults. Bermejo et al. [106] reported 5000 gorillas died of Ebola and, that is a lot of food biomass for future adult necrophagous and synanthropic flies. For gorillas and chimpanzees, it is calculated that 0.1–1% of the fresh weight of daily consumption is insects [107]. Their paper is a must read if one wants to better understand the role of insects in the diets of NHP. Using “DNA-barcoding primers that targeted the Cyt-b49 and COI50 genes in arthropod mitochondrial genomes”, they analyzed 9 fecal samples from gorillas, chimpanzees, and bonobos. Important dipterans found in the diets included two house flies each for the bonobo and gorilla assays and 1 house fly in the diet of one gorilla. Many other dipteran species were found and, if interested, one should consult their paper. Gogarten et al. [27] also raised the question as to whether pathogen transmission could result from primates either consuming flies or foods contaminated with their ‘fly vomit spots’. The major point here is that there is now sufficient evidence, using different techniques to sample feces of NHP, to conclude that NHP do eat insects and, if the flies eaten had crops full of pathogens, they could serve as infectious agents. An interesting, but appropriate, find was made by Qvarnström et al. [108] who reported finding many newly described and extinct beetles in a 230 MYO coprolite of a small dinosaur *Silesaurus opolensis*. Insects, including flies, have been around for a long time and have used feces of NHP, other animals, and hominids as a food source. Is it possible to find synanthropic fly bodies in coprolites of early hominids and NHP? 

The question that arises concerning NHP eating insects that might contain pathogens, either obtained from their body surface, from the crop, or from the midgut, is whether the viruses can make their way to the respiratory system without encountering the digestive system. This is an important question since humans [109] and primates both contain antimicrobial peptides that could destroy the virus or other microbes prior to entering the digestive tract where they face major destructive elements. Furthermore, the presence of antimicrobial peptides could destroy potential pathogens, which would prevent fly transmission. 

## 13. Do Flies Feed on the Feces and Other Fluids, such as Urine, of Gorillas, Chimpanzees, Monkeys? 

In general, the feces of most animals contain about 75% water, plus other ingredients. Insects, however, especially adult flies, must conserve water, their feces are usually deficient in water, and they must replenish water supplies by drinking. This need for water may help explain why various fly species are attracted to the eye and nasal secretions of humans and domestic animals [91]. We know little about flies being attracted to eye and nasal secretions of wild animals, especially gorillas and chimpanzees in the wild. 

For NHP, their feces contain various proportions of sterols (i.e., phytosterols and cholesterol) and stanols, which are excreted by wild chimpanzees (*Pan troglodytes*) and the mountain gorilla (*Gorilla beringei*) and can serve as fecal biomarkers for possibly providing new information about the palaeodiet of our early human ancestor [43]. As previously stated, flies, as other insects, cannot produce their own sterols. Thus, like some other animals, they rely on an exogenous source from plants and/or animals. These sterols are especially important for the insect because they use them to make essential hormones. Where adult flies obtained their sterols prior to the origin of carnivory has not been studied but, plant sterols were certainly available. We have discussed above that flies, which include previtellogenic females, plus males are attracted to most types of feces. They feed on them, regardless of the protein content, which can vary significantly depending on the animal and will have a subsequent effect on egg development [46]. Maybe, therefore female flies must also feed on more protein rich sources such as that obtained by carcass fluids. Few entomologists have really analyzed what flies are obtaining from vertebrate feces but, surely water and sterols would be acquired. More research is needed to explain what nutrients, especially sterols, flies obtain by feeding on feces. Current molecular technology [110] has permitted researchers to analyze many questions relating to environmental issues by using fecal samples, including their use to rapidly evaluate vertebrate biodiversity using blowflies as biodiversity monitoring species [53]. Both studies evaluate feces, while to my knowledge, these techniques have not been developed for analyzing urine. Figure 4. shows numerous flies attracted to the feces of a chimpanzee in Africa.

## 14. Do Flies Feed on Dead Non-Human Primates and Other Zoonotic Animals? 

Salvador Dali is one of the best examples of an entomophobe. He used ants, grasshoppers and flies in his paintings based on his personal experiences. In an analysis of his painting, The First Day of Spring, 1929, Lubar [111] writes, “…the image of a putrescent goat whose analogue is to be found in the swarm of flies that emerges from a woman’s head…” I am sure that many of you readers have experienced the swarms of flies at your food, on dead carcasses, usually along the roadside where they are feeding on dead animals and laying eggs. A common human behavior is to avoid flies, and, in addition to waving them away with our hands, we also use a fly swatter. There is now even reference that monkeys exhibit attempts to remove nuisance, bothersome flies by waving them away [67]. One can just imagine what it was, and still is like in various places like tropical Africa when NHP, such as a gorilla dies. Can you imagine what it would be like if flies could think about what a feast for their larvae is like having 5000 gorillas reported dying. In an interview with Dr. Karesh, the New York Times writer Donald G. McNeil, Jr. reported (14 October 2019), “The worst part of sampling a dead gorilla for Ebola, said Dr. William B. Karesh, is the flies. You can imagine the sense of panic,” he said. “A hundred thousand ants and carrion flies are coming off the carcass or climbing up your arms. They get inside your hood and are crawling on your face or biting you”.

We need more information on the number of dead, NHP carcasses, as well as other animals, present in tropical African forests, which should include other parts of the world, so that we can predict fly population densities. This will provide data on the availability of the carcasses for fly feeding and breeding. If flies are to be linked to diseases and, placed in epidemiological models, we need more information on their breeding sources in nature. We also need more information on the behaviors of NHP when they encounter or continually carry around dead infants. It is difficult to imagine that after dying, but still in their mothers’ arms, that flies are not attracted to the corpse for feeding and ovipositioning. In his paper on death in chimpanzees, Anderson [112] reports that, in one area of Tanzania, about half of the infants die before being weaned. How many bodies that entails, remains unknown, but to the flies it probably represents a significant and ever presence of adult food and larval sustenance. One of the main causes of death amongst chimpanzees is diseases (e.g., pneumonia, human respiratory viruses, simian immunodeficiency viruses, Ebola, and anthrax) [112]. Andersons paper mainly focuses on the socio-cultural aspects of chimpanzees that die, but he does discuss that chimpanzees when encountering a dead member of the group will make physical contact with the body and, in some instances eat some of the meat. Thus, contact with chimpanzees that have died from infectious diseases may present an infectious source to others, including flies, when they are touched and/or eaten.

In their paper, Porter et al. [113] discuss behaviors of *Gorilla beringei* spp., which include grooming, sniffing, and touching conspecific carcasses and include a section entitled, “Implications for disease transmission within and between gorilla social groups.” They raise concerns that NHP contact with dead conspecifics could lead to the transmission of numerous pathogens, including the Ebola virus. They note, “…final stage Ebola victims often secrete large quantities of infected bodily fluids, which could contaminate the surrounding ground and vegetation through infected saliva on discarded food remains and/or infected urine and feces left at the site”. Flies attracted to these carcasses surely would feed on these items and, as soon as their crops are full, they would fly to vegetation near the carcass and regurgitate any acquired pathogens that could then be picked up by other animals (e.g., as in the case of the anthrax example, see [94]. Unfortunately, nothing was said in their paper about the presence of flies at the sites of the dead gorillas. 

Today, there is an increase in the number of published papers devoted to forensics, mainly involving synanthropic flies [30]. We need to take many of these studies, the approaches they have developed, and apply them to epidemiological research on NHP and other animals in various tropical areas.

## 15. Do Bats Feed on these 3 Dipteran Families or the Fruit These Flies Regurgitated or Defecated Upon? How Long Can Pathogens from Fly Regurgitate or Defecation Remain Infective?

In a major review, Plowright et al. [114] discuss how various indirect transmission routes are involved in shed viruses in bat feces or urine and how they make their way into new hosts. Absent from what he calls “indirect transmission” routes are flies. Information on flies feeding on bat feces or urine is lacking. There are various reports that bats have fed on the 3 synanthropic groups discussed here. Rydell et al. [115] reported that members of all three families were collected in stomach analysis of 18 bats and from 4 species of bats which, researchers should be able to locate using echolocation. They noted that flies on a smooth surface should be easily located and eaten by the bats. This suggests that flies on a carcass of either elephants, gorillas, or hippopotamus (i.e., all of which are enormous supplies of food for hungry fly larvae), which two of them have rather smooth bodies, can be located and eaten, just like those flies on the windmill towers, as reported [116]. Information is needed as to whether flies in these 3 groups stay with a dead carcass overnight. If so, this might explain the above results where the investigators found flies, which were detected and eaten by bats on the smooth surfaces of windmills.

It has been demonstrated that non-enveloped viruses survive for longer periods of time, depending on the surface [116], while enveloped viruses can remain infectious for several days [117]. The study of Bean et al. [116] noted that the viruses (i.e., A and B virus) could survive for up to about 1 day depending on the surface. Some microbes like, *Mycobacterium bovis* are reported to survive more than 2 months while for *Bacillus anthracis*, as well as ORF virus [91], which can survive for several years in the soil. The statement by Fasanella et al. [39] that “Defibrinated rabbit blood positively influenced the replication of anthrax spores compared to the other substrata,” suggest that the type of substrate the pathogen is naturally located in when consumed and, taken to the crop storage organ, will also facilitate survival and possibly genetic modification. The laboratory studies by Brewer et al. [38] and Raele et al. [91] demonstrated that within the crop of adult house fly, the pathogens (*Chlamydia trachomatis* and the ORF virus) survived for 12 hrs and 6 hrs post-feeding, respectively. Of the discussed studies, the only one that looked at biofilm formation was Wang et al. [79]; however, the strategy of biofilm formation as a survival strategy for pathogens exists, not only for bacteria, but also fungi. Using the crop vessel assay [79], studies can be done using the normal media imbibed by the fly to examine factors influencing biofilm formation, bacterial quorum sensing factors for biofilm formation, and, also whether resistance or genetic reorganization takes place within the crop system studied.

More studies are needed on the length of survival, type of strategy for survival, and infectivity of various pathogens, especially those where flies are known to regurgitate and/or defecate on the food source or vegetation that is later consumed, as was done in the study by Wasala et al. [118]. They showed that adult house flies having access to *E. coli* from cow manure regurgitated on spinach and, the bacteria were still infectious after 13 days. The authors implicated mechanical transmission in their time studies but, it probably was obtained from the crop fluid that was then regurgitated, gotten onto the proboscis, and was spread throughout the fly’s surface via grooming. Bennett et al. [16] is a must-read where the authors found diverse arthropod origin viruses in the blood of the hammer-headed fruit Fanbats in the Republic of Congo. This paper provides a wealth of information concerning bats and viruses, which includes reference to bats consuming arthropods and bats possibly becoming infected via the gastrointestinal route. Caution must be taken as to the route of infection, because of the close morphological connection between the gastrointestinal route and the respiratory pathway. Few studies exist concerning these two closely connected systems and the importance of a host becoming infected via the respiratory route is a possibility and may provide less defenses than the gastrointestinal route. 

## 16. Examples from the Literature Supporting Flies as Extremely Likely Flying Capsules of Pathogens

Anthrax is considered one of the most important zoonotic emerging infectious diseases mainly affecting herbivores, which often involves fly vectors. In Kruger National Park, anthrax is endemic; and, as Basson et al. [94] note, its epidemiology often includes various fly transmitters of pathogens. They reported collecting 57 adult blowflies (*Chrysomya albiceps*, *Ch. marginalis* and *Lucilia* spp.) from a dead kudu and examining them for *Bacillus anthracis*. They found 65.5% of the flies contained only endospores and had fed on the kudu while 25% had fed on an infected impala carcass. In addition to their potential to transmit pathogens, many blow flies can greatly extend their range and disperse over large areas. A paper often overlooked on synanthrophic fly dispersal is that of Jones et al. [119]. As Maldonado and Centeno [47] reported, *C. albiceps*, originally found in Africa, is very common in Mediterranean regions, but is now distributed in Colombia, Argentina, Peru, and Paraguay where it is projected to be a major sanitary problem and involved in important emerging infectious diseases. 

Even though it is reported that these blow flies can travel long distances (35 km–65 km), once they have fed to repletion on infected bodily fluids, their crops and midguts become distended, the flies stop feeding, and they fly short distances (2.5 m) according to Basson et al. [94] from the dead body where they remain “bubbling” or regurgitating crop contents, which are often dropped onto the substrate they landed upon. In addition, they may defecate. This bubbling is essential to remove the water from the meal, lessen the body weight caused by the water, thus preparing the fly for efficient flight [26]. In the process, they often contaminate vegetation. Basson et al. [94] found defecation or regurgitation spots on the leaves of a shrub *A. angulatum* and stem of an Acacia tree. An earlier study by Fasanella et al. [39] let the common house fly (*M. domestica*) feed on *B. anthracis*—infected rabbit carcasses or *B. anthracis*—contaminated blood. They found the “…presence of *B. anthracis* spores in defaecation and vomit spots as well as hypothesized that anthrax spores are able to germinate and replicate in the gut (sic, of the fly)…” Neither study specifically identified the crop as a major storage organ for the pathogen. Basson’s group [94] concluded that more studies need to be done to determine if flies can carry infectious doses, whether they deposit infectious doses via their excreta, and whether these contaminated droplets, presumably via regurgitation, on vegetation are enough to infect another herbivore. These suggestions need to be addressed to place blowflies as part of the epidemiological system of anthrax.

In addition to the African anthrax situation in Kruger National Park, a study was done in West Texas where outbreaks of anthrax were reported in wildlife; and an attempt was made to confirm the link in the anthrax-animal-fly model [120]. The initial study [103], which was the first field study in the U.S., confirmed that necrophilic flies collected during the 2005 anthrax season in West Texas carried spores of the bacterium. These researchers [120] used PCR, plus classical microbiological assays to demonstrate the presence of the bacterium in adult flies and in the vegetation near the animal that died of anthrax infection. Their results confirmed in the field that the animal-to-plant transmission cycle involves blowflies of the Calliphoridae and Sarcophagidae families feeding on the fluids of the dead animal. Once engorged, they rest on vegetation and deposit contaminated droplets as either feces and/or regurgitant. Thus, the anthrax-infected animal-surrounding vegetation epidemiological model is facilitated, multiplied, and enhanced by non-blood feeding flies feeding on the infected carcass.

## 17. The Immune System of Flies and Other Vertebrates

In 2011, Julius Hoffman received and shared the Noble Prize in Physiology or Medicine for his research on the activation of innate immunity in the fly workhorse of biology, *Drosophila melanogaster*. It is not possible to do justice to this topic in this review with respect to all the synanthropic flies other than to mention a few studies and areas where more research is needed if we are to understand the importance of these flies as transmitters of important pathogens of humans, domestic, and wild animals. The importance of antimicrobial peptides present in the saliva and secretions produced by the salivary and labellar glands of flies, plus those in the midgut, need to be carefully researched to better understand how flies can act as efficient transmitters of pathogens [51] and to better understand how their defensive system avoids infection. 

The first major, internal encounter pathogens have with a potential vector is within the foregut where the crop AMPs from the salivary and labellar glands have deposited their secretions, along with the food and pathogens imbibed and eventually may enter the midgut [121]. A major review on the role of the fly foregut, which includes the crop, with respect to microbes is provided [51]. One can search the literature to find appropriate papers that pertain to the synanthropic flies, the type of AMPs present, and what effect they might have on pathogen success. A most interesting paper is that of Schlein who called the adult crop a ‘sterilization organ’, without which the leishmanial promastigotes could not survive [51]). No one has identified the AMPs in this fly and determined how effective they are against pathogens imbibed with their food (e.g., in this case carbohydrates from nectar). For flies to be effective in and participating in any epidemiological model of emerging infectious diseases, information about their immune systems is critical. 

## 18. Conclusions

Future studies aimed at showing the involvement of the flies discussed in this study should be specific in locating what part of the digestive tract was studied. This will avoid individuals, such as in the study by Calvignac-Spencer [122,123] referring to the fly’s digestive tract as a stomach. 

Individuals should realize that globalization does not only refer to economics, as so aptly reported by Friedman [124]. We not only share the world with other people, but all life forms are included in this flat, interconnected world so aptly presented by Friedman [124]. As the world’s population increases, humans expand agricultural systems to feed everyone. Oftentimes this involves destruction of specific habitats, like forests, that place us in direct contact with wild reservoir hosts that are ready to share their pathogens via flies. 

The paper by Rose et al. [125] presents information about how climate change and urbanization have driven mosquito preference for humans. This is an excellent example of determining a behavioral or physiological mechanism connecting a mosquito with humans. To date, the only example of connecting synanthropic flies with humans is the house fly lactose connection and, this needs m ore research. What impact climate change will have on synanthropic flies has not been addressed. How urbanization impacts the dynamics of the wildlife-livestock-human interface was addressed and hopefully synanthropic flies will play an important role within this dynamic interface [126]. Is humanity ready to seriously take on the responsibility of continued surveillance for emerging infectious diseases? If so, the information presented in this review, the importance of cyclorrhaphan flies in epidemiological models, using the newest techniques to examine these biosentinel flies for potential hazards, and pathogen transmissions is vital.

Synanthropic flies may be even more important in disease transmission than blood-sucking flies and, the role of the crop in proving this should help. Future research will also help answer this question.

Finally, a schematic diagram (Figure 5) shows the involvement of synanthropic flies as transmission agents of diverse pathogens and their role in an epidemiological model. 

Synanthropic flies ingesting pathogens from feces, secretions from infected animals, or infected foods. These liquids go into the crop where they remain and/or resistant forms are generated. Following a large meal and, when the crop is full, they either regurgitate crop fluids, which now contains infective pathogens, or the liquids can be moved into the rest of the digestive tract to infect various sources via defecation. Blue stars are various pathogens while the odd, red-shaped objects are pathogens that have become antimicrobial resistant by various strategies within the crop but are then transferred by the fly for uptake by uninfected hosts. Figure 5A. In this figure section, an infected monkey, and bat release in their feces or urine while shedding various pathogens, especially in their feces that are very attractive to flies. The flies imbibe the liquids containing the pathogens, which go to the storage organ (i.e., the crop). Sometimes, bats eat, but not completely, fruits that they infect with their saliva, which are then dropped and can be picked up by flies or other animals such as duikers on the ground to become infected. Figure 5B. Flies are attracted to foods, such as mangoes and feces where they pick up pathogens, or resistant pathogens, which go into the crop and are deposited elsewhere. These pathogens are released to another source either via regurgitation, thus contaminating various food sources or contaminate, via mechanical transmission or by regurgitation (bubbling). Figure 5C. Flies feed on the various solutions from dead animals (i.e., dead chimpanzee, bottom right) and put the solutions into the crop, which later the fly regurgitates or defecates the pathogens onto foods such as bushmeat (top right) in the wet markets. Flies can also pick up pathogens, such as anthrax spores by eating dead, infected animals (bottom right). Humans become infected with pathogens or resistant pathogens by eating contaminated fruit from either fecal deposition or regurgitation by infected flies (Figure 5D).

## Figures and Tables

**Figure 1 insects-13-00776-f001:**
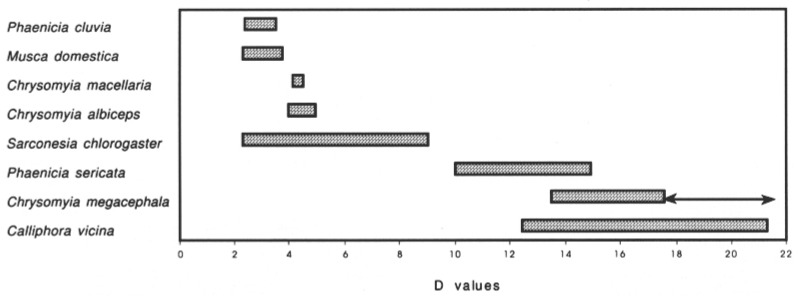
Taken from Maldonado and Centeno [47] showing the danger index (D) for flies collected and the arrow line showing the changes for *Chrysomyia megacephala* as considered now as hemisynanthropic (D = 19.43).

**Figure 2 insects-13-00776-f002:**
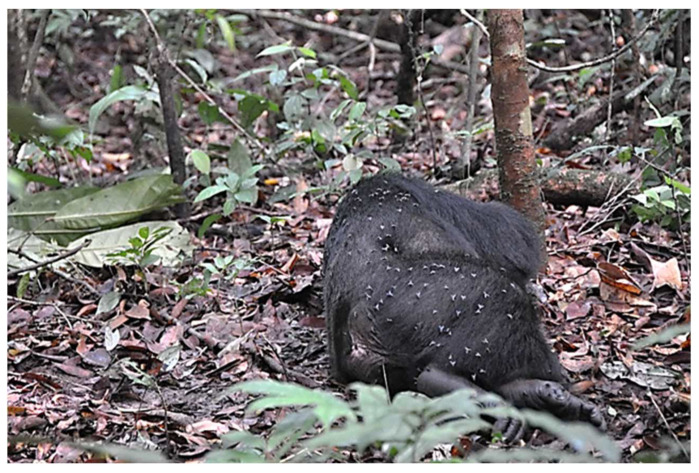
Flies attracted to a sleeping chimpanzee. Photo courtesy of Fabian Leendeertz and the Tai Chimpanzee Project.

**Figure 3 insects-13-00776-f003:**
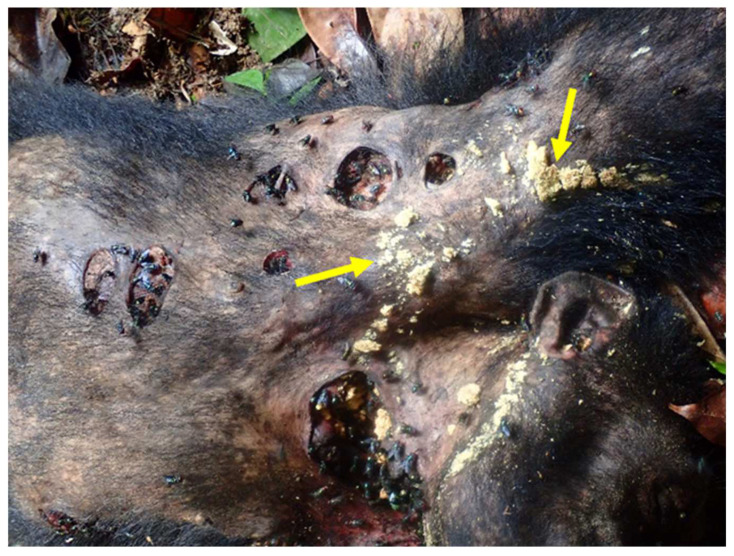
Flies feeding and laying eggs (arrows) on a dead chimpanzee. Photo courtesy of Jennifer Jaffee and the Tai Chimpanzee Project.

**Figure 4 insects-13-00776-f004:**
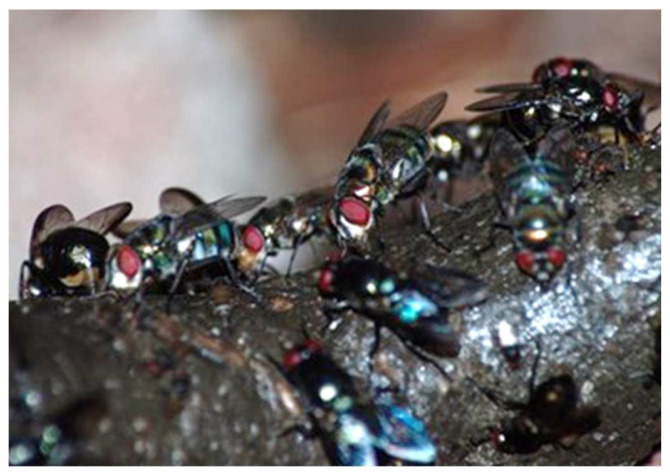
Flies feeding on the feces of a mangabey. Photo.courtesy of Jan Gogarten and the Tai Chimpanzee Project.

**Figure 5 insects-13-00776-f005:**
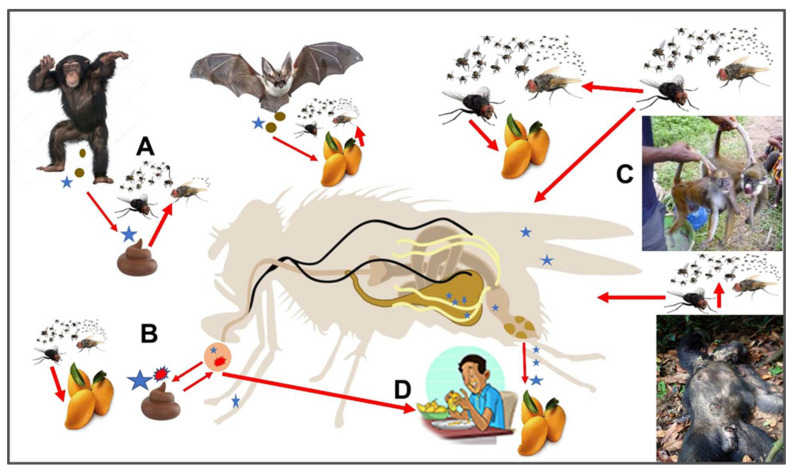
The fly image is the courtesy of Dana Nayduch and is modified from Nayduch and Burrus, [29], 2017, while the dead chimpanzee photo is the courtesy of Jennifer Jaffee and the Tai Chimpanzee Project. A more detailed explanation is provide for the various subfigures (**A**–**D**). Drawing of mango eater by Sahil Upalekar.

## Data Availability

Not applicable for studies not involving humans or animals.

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
