# Peer review of "Synanthropic Flies—A Review Including How They Obtain Nutrients, along with Pathogens, Store Them in the Crop and Mechanisms of Transmission"

_insects, 2022, doi:10.3390/insects13090776_

Round 1
Reviewer 1 Report
I thoroughly enjoyed reading this very informative and well-written review. My comments are nearly entirely of an editorial nature and follow:
Line 23 move comma to after “insects”
Lines 129-132. This is a very long parenthetical remark. Sentence should be re-arranged
Line 142 and throughout the manuscript - Scientific names should be italicized
Line 144 The language “to convince you” is jarring, almost aggressive. Suggest softening to “make the case that”
Line 163, change to hosts
Line 185 remove first comma
Line 186 change into to in
Line 198 remove comma after Did
Line 234 remove comma before include
Line 237 remove “but badly needed”
Line 257 change “a lot of” to many
Line 271 and throughout - “dead carcasses”. The most common usage of carcass is that of a dead animal. In my opinion the “dead” is redundant.
Line 311 change surprising to surprisingly
Lines 329-338 tells an interesting personal anecdote that is not necessary
Line 348 remove comma after but
Line 386. I don’t agree that house flies “ignore the nectar of flowers”. They are often found feeding on nectar of flowering plants around stables and barns.
Line 390. Do you mean “I propose that”?
Line 426. Do you mean asking instead of aging?
Line 469 sentence should not end with a question mark
Line 473 delete comma after done
Line 534 remove comma after eggs
Line 572 wording (in OUR now classical) suggests that the author did the work
Lines 581-582. Do we really know this? It depends on the amount of inoculum needed to infect the animal. Soften the “surely” since we don’t know.
Line646 change infected to contaminated
Lines 684-685 change to They found that
Line 822 delete comma after “Can you imagine”
Line 905 change virus was to bacteria were (if you’re still talking about E. coli)
Line 935 right paren after [96] (I think)
Line 958 change there to their
line 997 delete comma after wild
Line 1037 add Project after Chimpanzee
Author Response
Thank you for comments and suggestions. I have carefully made the necessary adjustments on complex sentence structure. I disagree with your suggestions about the paragraph involving Dr. Stephen Rich. As author I would like to include this paragraph in the paper.
With respect to the length, this is a major review and I feel the length is warranted.
Reviewer 2 Report
The author presents useful information on an important topic, viz., the role of synanthropic flies in vector-borne disease. The author has clearly given much thought to this topic. However, the manuscript is laborious to read. It is long, wordy, and often redundant. Consequently, the central message tends to be lost. I strongly urge the author to rework the manuscript, with an eye toward clarity, brevity, and precision to allow readers to focus on the main points.
The title is long and cumbersome. A shorter, more manageable title would be more attractive, for example: “Synanthropic Diptera (Calliphoridae, Sarcophagidae, and Muscidae) as Transport Vehicles in the Transmission of Primate Pathogens and Parasites”.
Typos and poorly constructed sentences create confusion throughout the manuscript. A few examples follow, but please note that these are only a few of the many examples:
- Run-on sentences (e.g., lines 163-165).
- Repetition (e.g., lines 176-177: “The animal that is reported to be… is reported to be”).
- Confusing constructions (e.g. line 200: “When imbibed, these adult flies take the eye secretions”).
- Incomplete sentences (e.g., lines 204-205: “Surely enough time to infect, via regurgitation, a new host.” And line 385: “Thus, a new food source for adult house flies.”).
- Needles verbiage (e.g., line 323: “It is significant to note that in their extensive review”; this phrase could easily be deleted without losing the author’s point).
- Careless wording (e.g., line 829: “dead, NHP carcasses”; presumably all carcasses are dead; and line 870: “either bat feces and/or urine”—only “or” matches with “either”; “and” does not go with “either”).
- Misplaced or misused punctuation, especially commas.
- Misused apostrophes (e.g., line 671: “larger fly’s better transmitters of pathogens”)
- Scientific names need to be italicized throughout the manuscript.
- The imprecise use of the slash should be avoided (as in studies/models, parasites/pathogens, microbes/parasites, necrophagous/synanthropic, regurgitation/defecation, fly/monkey, hours/days, in/on, human/domestic/wild, food/pathogens, etc.). Please use more precision to indicate what is meant by the slash (e.g., whether the slash means “and”, “or”, or something else). Presumably, the slash means something very different for “parasites/pathogens” than for “fly/monkey”).
The word “vector” in the context of the manuscript does not exist as a verb. Thus, “vectoring” (lines 11, 316, 428, 505) is not appropriate; “transmitting pathogens” is sufficient.
Lines 171, 365, etc.: The term “primatothropic” seems to be a neologism. If so, it should be removed. The word is malformed, combining the Latin “primat-“ meaning ‘first rank’ with the Greek “anthropos” meaning ‘man’. It is, therefore, an inappropriate combination of Latin and Greek, and it has the wrong literal meaning.
Lines 211-212: The citation “Benfield and others 1992” does not appear in the References and it is not clear how it relates to reference [40]. Please clarify attribution of the quote, or better, put the quote into the author’s own words.
Lines 252-253 (“As some NHP groups face extinction, what effect will this have on the flies?”): It is not entirely clear what point the author wishes to make here. Is it to imply that the flies are species-specific in their choice of host carcasses or dung? If so, an example would be helpful. The families of flies treated by the author tend to be highly opportunistic in their choice of feeding and breeding habitats so that extinction of a particular host species would presumably have little effect on a generalist or opportunistic species of fly.
The excessive use of quotes seems unnecessary. Nearly all of the quotes would be better presented in the author’s own words.
Lines 569-570 (“The crop is an isolated foregut, diverticulated storage organ, unique to the Diptera…”): The crop, whether as a diverticulation or not, is not unique to Diptera. Most insects have a crop and in some groups additional to Diptera, such as the higher Lepidoptera, the crop is a diverticulation. Additionally, the crop is a part of the foregut, rather than “an isolated foregut”.
Lines 994-995 (“all life forms are included in this flat world”). What is meant by “flat world”?
Author Response
Thank you for your comments and suggestions that improved the paper.
Reviewer 3 Report
Dear Editor and Authors,
The review "Significance of the Synanthropic Dipteran Families, Calliphoridae, Sarcophagidae, and Muscidae, as Flying Transport or Stor-3 age Vehicles Involved in the Transmission of Important, Non-4 Human Primate and Human Pathogens/Parasites”
Presents detailed summaries of literature from a variety of sources covering broad topics related to the association between insects and primates. The potential of insects act as vectors of disease and the potential function of the crop in disease transmission by flies.
Unfortunately, this broad approach taken in this manuscript means that the core aims of the review are obscured and none of the individual topics are covered in a focused and concise manner.
There are aspects of the paper that are very good, and I have highlighted them in the text, but these are outweighed by sections that do not fit in with the scope of the review as outlined int the title, abstract and summary. Many entire sections could be removed – which would result in a more focused manuscript. There are also sections which I do not understand the rational of including (i.e. the paper generally focuses on transfer of disease from insects to non-human primates and humans, and yet there are several sections which deal specifically with bats – but no other wildlife).
There are many bold claims in the text which are stated without references and based on assumptions which I believe to be incorrect – I have indicated there in the main text along with other specific comments.
While there are interesting aspects to this paper, I cannot recommend its publication at this time. I believe the paper is unfocused and would benefit from extensive restructuring, I encourage the author to choose a more focused aim for this review and cover it more thoroughly. To this end I believe the topics covered paper could easily form the basis of 2-3 separate reviews if each were covered in sufficient detail.

Round 2
Reviewer 2 Report
Please see the following comments:
The title is grammatically problematic. Better: "Synanthropic Flies – A Review of How They Obtain and Store Nutrients and Transmit the Acquired Pathogens"
Line 1054: reference to a "flat world" must be a mistake or, if it is intended as a literary device, it should be removed.
Lines 1072-1074: Best to remove the statement that synanthropic flies are more important to transmission of disease agents than are blood-sucking flies. This is an apple-orange comparison. In most cases, the pathogens and parasites transmitted by the two groups of flies are different, as are the resulting diseases. This opinion would be difficult to "prove" in any case. I would delete the entire paragraph.
Author Response
I have considered the review comments and have usually incorporated them. Thanks to them.
Reviewer 3 Report
Dear Editor and Authors,
The review "Significance of the Synanthropic Dipteran Families, Calliphoridae, Sarcophagidae, and Muscidae, as Flying Transport or Stor-3 age Vehicles Involved in the Transmission of Important, Non-4 Human Primate and Human Pathogens/Parasites”
The author presents detailed summaries of literature from a variety of sources covering a very broad range of topics related to the association between insects and primates. The potential of insects to act as vectors of disease and the potential function of the crop in disease transmission by flies.
While the author has taken steps to improve the manuscript – many of the comments I originally provided have not been addressed and while some are my opinions on what would improve the manuscript others point towards important omissions of information that may be misleading to the reader. The broad approach taken still means that the core aims of the review are obscured and none of the individual topics are covered in a focused and concise manner. Many entire sections could still be removed and there are still sections which I do not understand the rational of including. There are still many bold claims in the text which are stated without references and based on assumptions which I believe to be incorrect.
Based on these reasons I cannot recommend its publication at this time.
Author Response
I agree with most of your comments and have incorporated them
Thank you